# GLIF: A Unified Gated Leaky Integrate-and-Fire Neuron for Spiking Neural Networks

**Xingting Yao**[1,2], **Fanrong Li**[1,2], **Zitao Mo**[1], **Jian Cheng**[1,3*]

[1]Institute of Automation, Chinese Academy of Sciences
[2]School of Future Technology, University of Chinese Academy of Sciences
[3]CAS Center for Excellence in Brain Science and Intelligence Technology
{yaoxingting2020, lifanrong2017, mozitao2017, jian.cheng}@ia.ac.cn,

## Abstract

Spiking Neural Networks (SNNs) have been studied over decades to incorporate their biological plausibility and leverage their promising energy efficiency. Throughout existing SNNs, the leaky integrate-and-fire (LIF) model is commonly adopted to formulate the spiking neuron and evolves into numerous variants with different biological features. However, most LIF-based neurons support only single biological feature in different neuronal behaviors, limiting their expressiveness and neuronal dynamic diversity. In this paper, we propose GLIF, a unified spiking neuron, to fuse different bio-features in different neuronal behaviors, enlarging the representation space of spiking neurons. In GLIF, gating factors, which are exploited to determine the proportion of the fused bio-features, are learnable during training. Combining all learnable membrane-related parameters, our method can make spiking neurons different and constantly changing, thus increasing the heterogeneity and adaptivity of spiking neurons. Extensive experiments on a variety of datasets demonstrate that our method obtains superior performance compared with other SNNs by simply changing their neuronal formulations to GLIF. In particular, we train a spiking ResNet-19 with GLIF and achieve $77.35\%$ top-1 accuracy with six time steps on CIFAR-100, which has advanced the state-of-the-art. Codes are available at `https://github.com/Ikarosy/Gated-LIF`.

## 1 Introduction

Spiking neural network (SNN) is considered the third generation of the neural network, and the bionic modeling of its activation unit is called the spiking neuron, which brings about the spike-based communication between layers [1, 2]. Due to the high sparsity and multiplication-free operation in information processing, such spike-based communication can improve energy efficiency [3, 4]. Moreover, spiking neuron can effectively capture temporal information and has good performance in processing neuromorphic data [5] and anti-noise [6]. Together with artificial neural network (ANN) colleagues, SNN further shows great potential for general intelligence [7]. Thus, SNN has aroused heated research interests recently.

The commonly used spiking neuron in different SNNs [6, 8, 9, 10, 11, 12] is the leaky integrate-and-fire (LIF) model, named vanilla LIF. As the name suggests, it can implement three different neuronal behaviors: membrane potential leakage, integration accumulation, and spike initiation. To better model neurons, prior works [13, 14, 15, 16, 17] have improved the vanilla LIF from the above three aspects, and proposed many LIF variants with different biological features. For all these vanilla and variant LIF models (referred to as simplex LIF), any of the three neuronal behaviors supports

---

*Corresponding author.

36th Conference on Neural Information Processing Systems (NeurIPS 2022).

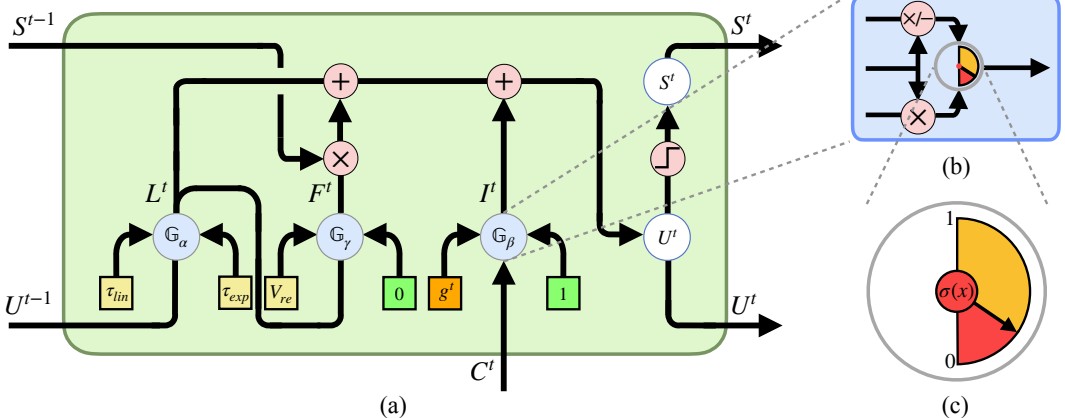

Figure 1: Illustration of the GLIF model. (a) Discrete GLIF neuron. The membrane potential $U^t$ and the output spike $S^t$ are updated over discrete time steps. Gating units $\mathbb{G}_\alpha$, $\mathbb{G}_\beta$, and $\mathbb{G}_\gamma$ fuse dual bio-features in membrane potential leakage, integration accumulation, and spike initiation, respectively. (b) Gating unit. (c) Gating factor.

only single biological feature. For instance, the vanilla LIF model only utilizes exponential decay in membrane potential leakage, while [15] only supports linear decay. However, different neurons in the cerebral cortex have different response characteristics [18, 19].This naturally raises an issue: *could the higher neuronal dynamic diversity of spiking neurons help SNNs do better?* In this paper, we investigate the hybrid bio-features in the LIF model to answer this question.

We propose the gated LIF model (GLIF) that fuses different bio-features in the aforementioned three neuronal behaviors to possess more response characteristics. As illustrated in Fig. 1, GLIF controls the fusion of different bio-features through gating units $G_\alpha$, $G_\beta$, and $G_\gamma$ for those three neuronal behaviors that are membrane potential leakage, integration accumulation, and spike initiation, respectively. In each gating unit, a gating factor is computed from a Sigmoid function $\sigma(x)$ over a learnable gating parameter $x$ to determine the proportion of each bio-feature, thus guiding the fusion of different bio-features. Therefore, GLIF can simultaneously contain different bio-features, possessing more response characteristics. In addition, when the gating factor is $0$ or $1$, GLIF can also support single bio-feature. As a result, GLIF can cover other different LIF models and be viewed as a super spiking neuron, greatly enlarging the representation space of spiking neurons.

Furthermore, we introduce the channel-wise parametric method to GLIF. This method makes all membrane-related parameters in GLIF learnable and shares the same GLIF parameters channel-wisely in SNNs. Combining with learnable gating factors in GLIF, on the one hand, this method makes different channels in SNNs have completely different spiking neurons, leveraging the larger representation space of GLIF neurons to increase the neuronal dynamic diversity of spiking neurons. Meanwhile, the heterogeneity of spiking neurons and the expressive ability of SNNs are also increased. On the other hand, the spike neurons in SNNs are constantly changing during training, which is similar to the neuronal maturation during development [20, 21, 22], thus the adaptivity of spiking neurons being enhanced.

Our contributions are as follows:

- We propose the GLIF model, a unified formulation of the spiking neuron that fuses multiple biological features through gating units, enlarging the representation space of spiking neurons.

- We exploit the channel-wise parametric method to leverage the larger representation space of GLIF, thus increasing the neuronal dynamic diversity in SNNs.

- Our experiments demonstrate the effectiveness of the GLIF model on both static datasets and the neuromorphic dataset. For example, our model can achieve the state-of-the-art $75.48\%$ top-1 accuracy on CIFAR-10 with only two time steps.

## 2 Related Work

**Variant LIF.** Variant LIF models with different biological features are proposed in previous arts, and those models differ in one or more of the three different neuronal behaviors: membrane potential leakage, integration accumulation, and spike initiation. Firstly, membrane potential leakage represents the potential decay mechanism. Exponential decay is first exploited in the vanilla LIF model [23], and it brings neuronal dynamics with stable convergence, shrinking the potential to the resting potential steadily. But, exponential decay is not efficient enough in massive neural simulations due to its multiplication operation. Therefore, linear decay replaces exponential decay in [13, 24, 25, 26, 27], and brings neuronal dynamics with the flexibility, making it possible for configurable spiking rates [15]. Although the two types of decay behave differently in neuronal dynamics, both of them are proven effective in SNNs [6, 28]. Secondly, integration accumulation represents the potential accumulation induced by presynaptic spikes. Commonly, spike intensities of different time steps are considered equally high. Consequently, the discrepancies among the spikes at different time steps are ignored. To address this issue, many prior works propose to assign different weights to the input spikes at different time steps. These methods are implemented by manual settings [29, 30, 31] or adaptive settings [32, 33, 34] to improve biological plausibility, SNN performance, or inference efficiency. Lastly, spike initiation functions as the resetting mechanism triggered by spike firing. There are two common resetting strategies: hard reset [35] and soft reset [15, 36]. When a spike is triggered, the hard reset directly sets the potential to the resting potential, while the soft reset subtracts the potential by a configurable value. Therefore, hard reset is more stable as a strict reset mechanism, while soft reset is more flexible, making configurable spiking rates feasible [15]. Compared with the LIF models above, our GLIF incorporates different neuronal behavioral mechanisms to increase the representation space of LIF models and improve the performance of SNNs.

**Parametric spiking neurons.** Early existing spiking neurons often require manual tuning of the membrane-related parameters to set their neuronal dynamics, and these parameters are often shared by all neurons, which limits the diversity of spiking neurons and the expressive ability of SNNs. In recent years, parametric spiking neurons [37, 38, 10, 28] have been proposed, which set one or more membrane-related parameters to be learnable, and update these parameters during the training of SNNs. [37] and [38] set the membrane leak and potential threshold learnable, while [10] and [28] choose the membrane leak only. FS-neuron [39] is more aggressive, setting all membrane-related parameters in the spiking neuron learnable to strictly fit the neuronal dynamics to the target activation function. Compared with the above methods, GLIF adopts the channel-wise parametric method to fully parameterize the spiking neuron, including learnable exponential decay, linear decay, potential threshold, resetting voltage, input conductance, and gating factors.

**Supervised direct learning of SNNs.** Supervised direct learning of SNNs follows the idea of backpropagation (BP) in ANN [40]. Bohte *et al.* [41] first adopt the surrogate gradient (SG) to empirically solve the non-differentiable term of spiking neurons by approximation with the local derivative of potential. Based on the observation that the proper steepness of the SG curve matters more than the shape in SNN learning [42], Li *et al.* [11] conduct a pre-estimation before each epoch to find the SG curve with the optimal smoothness. Besides, many techniques are developed to make deep SNNs converge faster, such as NeuNorm [8], tdBN [9], SEW block [43], TET [12], MS residual block [44], etc. In this way, the problem of end-to-end training of deep SNNs can be solved. In this paper, we choose supervised direct learning of SNNs to estimate the effectiveness of our GLIF.

## 3 Methodology

### 3.1 Revisiting LIF models

**Vanilla LIF.** The Vanilla LIF model is commonly adopted as a spiking neuron model in SNNs, which can be discretely formulated as follows:

$$\mathbf{U}^{(t,l)} = \tau \mathbf{U}^{(t-1,l)} \odot (1 - \mathbf{S}^{(t-1,l)}) + \mathbf{C}^{(t,l)}, \tag{1}$$

$$\mathbf{C}^{(t,l)} = \mathbf{W} \cdot \mathbf{S}^{(t,l-1)}, \tag{2}$$

$$\mathbf{S}^{(t,l)} = \mathbb{H}(\mathbf{U}^{(t,l)} - V_{th}). \tag{3}$$

Here, $\odot$ denotes the element-wise multiplication. $\mathbf{U}^{(t,l)}$ is the membrane potential vector of the layer $l$ at time-step $t$ and can be updated in a discrete way through Eq.(1), where $\mathbf{C}^{(t,l)}$ denotes the input vector and can be obtained by the dot-product between the synaptic weight matrix $\mathbf{W}$ and the output spike vector of the previous layer $\mathbf{S}^{(t,l-1)}$, as described in Eq.(2). The output spike vector $\mathbf{S}^{(t,l)}$ is given by the Heaviside step function $\mathbb{H}(\cdot)$ in Eq.(3), indicating that a spike is fired when the membrane potential exceeds the potential threshold $V_{th}$. When updating $\mathbf{U}^{(t,l)}$ in Eq.(1), the three neuronal behaviors are considered. Specifically, the time constant $\tau$ acts as the exponential decay coefficient on the term $\tau \mathbf{U}^{(t-1,l)}$ to implement exponential decay, $\mathbf{C}^{(t,l)}$ is integrated over time to $\mathbf{U}^{(t,l)}$, and the term $(1 - \mathbf{S}^{(t-1,l)})$ hard resets the membrane potential to 0 if a spike is fired at the previous time step.

**Bio-features & Primitives.** Bio-features represent the characteristics of the three critical neuronal behavior. For the instance of vanilla LIF, its bio-features can be described as exponential decay, uniform-coding scheme, and hard reset. And, the three bio-features qualitatively describe the neuronal behavior of membrane potential leakage, integration accumulation, and spike initiation, respectively. Since each behavior is actually implemented by the corresponding term in formulas, for explicitness, we refer to those neuronal behavior-related parameters as **primitives** in this paper. In this way, bio-features can be quantitatively described by primitives. For example, exponential decay is realized by the term $\tau \mathbf{U}^{(t-1,l)}$ in Eq.(1), so $\tau$ is a primitive and determines the amplitude of exponential decay. Thus, by choosing different primitives for each behavior, we can formulate different LIF variants. For instance, with the linear decay primitive $\tau_{lin}$ and the soft reset primitive $\tau_{re}$, we have: $\mathbf{U}^{(t,l)} = \mathbf{U}^{(t-1,l)} - \tau_{lin} - V_{re}\mathbf{S}^{(t-1,l)} + \mathbf{C}^{(t,l)}$, which formulates the spiking neuron in [15].

## 3.2 Gated LIF Model

Although variant LIF offers a different choice in modeling the spiking neuron, it still remains simplex, i.e., the bio-features of the neuronal dynamic are unitary. Once formulated, a hard-reset LIF neuron cannot perform the soft-reset operation. As a result, these simplex LIF models make the spiking neurons in SNNs lack dynamic diversity. Thus, we propose the gated LIF (GLIF), a unified spiking neuron, to fuse different bio-features in the three neuronal behaviors. GLIF is illustrated in Fig. 1 and further described as follows:

$$\mathbf{U}^{(t,l)} = \mathbf{L}^{(t,l)} + \mathbf{I}^{(t,l)} + \mathbf{F}^{(t,l)} \odot \mathbf{S}^{(t-1,l)}, \tag{4}$$

$$\mathbf{C}^{(t,l)} = \mathbf{W} \cdot \mathbf{S}^{(t,l-1)}, \tag{5}$$

$$\mathbf{S}^{(t,l)} = \mathbb{H}(\mathbf{U}^{(t,l)} - V_{th}), \tag{6}$$

$$\mathbf{L}^{(t,l)} = \mathbb{G}_\alpha(\mathbf{U}^{(t-1,l)}; \tau_{lin}, \tau_{exp}), \tag{7}$$

$$\mathbf{I}^{(t,l)} = \mathbb{G}_\beta(\mathbf{C}^{(t,l)}; g^t, 1), \tag{8}$$

$$\mathbf{F}^{(t,l)} = \mathbb{G}_\gamma(\mathbf{L}^{(t,l)}_{exp}; V_{re}, 0). \tag{9}$$

In Eq.(4), $\mathbf{L}^{(t,l)}$ denotes the membrane potential vector after membrane potential leakage, $\mathbf{I}^{(t,l)}$ denotes the vector of the incremental potential caused by integration accumulation, and $\mathbf{F}^{(t,l)}$ denotes the vector of the potential reduction caused by spike initiation. Obviously, the above three vectors reflect the consequences of the three neuronal behaviors. In Eq.(7)(8)(9), the gating units $\mathbb{G}_\alpha(\cdot)$, $\mathbb{G}_\beta(\cdot)$, and $\mathbb{G}_\gamma(\cdot)$ are proposed to formulate the three neuronal behaviors, thus computing the consequent vectors.

**Computing $\mathbf{L}^{(t,l)}$.** In computing $\mathbf{L}^{(t,l)}$, the two primitives $\tau_{lin}$ and $\tau_{exp}$ are considered. $\tau_{lin}$ is the linear decay primitive, and $\tau_{exp}$ is the exponential decay primitive. As their names indicate, $\tau_{lin}$ and $\tau_{exp}$ respectively play the roles of the linear decay and exponential decay on $\mathbf{U}^{(t-1,l)}$. Through the gating unit $\mathbb{G}_\alpha(\cdot)$, the two primitives are fused together to perform membrane potential leakage. Following is the formulation of $\mathbb{G}_\alpha(\cdot)$:

$$\mathbb{G}_\alpha(\mathbf{U}^{(t-1,l)}; \tau_{lin}, \tau_{exp}) = [1 - \alpha(1 - \tau_{exp})]\mathbf{U}^{(t-1,l)} - (1 - \alpha)\tau_{lin}, \tag{10}$$

where $\alpha$ is the gating factor of $\mathbb{G}_\alpha(\cdot)$, and its value is bounded within $(0, 1)$. When $\alpha$ approaches 0, $[1 - \alpha(1 - \tau_{exp})]$ and $(1 - \alpha)$ tend to 1, which means GLIF performs linear decay. As $\alpha$ increases

from 0 to 1, $(1 - \alpha)\tau_{lin}$ decreases from $\tau_{lin}$ to 0, while $[1 - \alpha(1 - \tau_{exp})]$ decreases from 1 to $\tau_{exp}$, which means GLIF is shifting from linear decay to exponential decay. Therefore, in Eq.(10), the two primitives $\tau_{lin}$ and $\tau_{exp}$ are dually fused with the help of the gating factor $\alpha$, which can balance the amplitudes of linear decay and exponential decay. Note that the primitives in the same gating unit are referred to as the ***dual primitives*** in the following content, and correspondingly, the bio-features the dual primitives describe are referred to as the ***dual bio-features***. In the extreme case of $\alpha = 0$ or 1, $\mathbb{G}_\alpha$ preserves only single primitive, which means GLIF covers the simplex formulations of membrane potential leakage.

**Computing $\mathbf{I}^{(t,l)}$.** In computing $\mathbf{I}^{(t,l)}$, the dual primitives $g^t$ and *1* are considered to perform integration accumulation. $g^t$ denotes the conductance primitive, namely, the time-dependent synaptic weight, which can be coded with different patterns, such as rank-order based patterns [29] and phase-coding based patterns [31]. So, we can view it as the flexible-coding primitive. While, the constant *1* means $g^t \equiv 1$, representing the uniform-coding primitive, which is the dominant paradigm and proven doing well in the direct learning of SNNs [45, 12, 11]. Following is the formulation of $\mathbb{G}_\beta(\cdot)$:

$$\mathbb{G}_\beta(\mathbf{C}^{(t,l)}; g^t, 1) = [1 - \beta(1 - g^t)]\mathbf{C}^{(t,l)}, \tag{11}$$

where $\beta$, bounded within $(0, 1)$, is the gating factor of $\mathbb{G}_\beta(\cdot)$. Similarly, as $\beta \to 0$, the uniform-coding pattern is preferred rather than the flexible-coding, while $\beta \to 1$ is the opposite case. When $\beta = 0$ or $\beta = 1$, GLIF supports single spike coding pattern in integration accumulation.

**Computing $\mathbf{F}^{(t,l)}$.** In computing $\mathbf{F}^{(t,l)}$, the soft-reset primitive $V_{re}$ and the hard-reset primitive *0* are coupled as the dual primitives to perform spike initiation. Following is the formulation of $\mathbb{G}_\gamma(\cdot)$:

$$\mathbb{G}_\gamma(\mathbf{L}_{exp}^{(t,l)}; V_{re}, 0) = -(0 + \gamma)\mathbf{L}_{exp}^{(t,l)} - (1 - \gamma)V_{re}, \tag{12}$$

where $\gamma$, bounded within $(0, 1)$, is the gating factor of $\mathbb{G}_\gamma(\cdot)$. The term $(1 - \gamma)V_{re}$ is the actual potential decrease caused by soft reset, while the term $(0 + \gamma)\mathbf{L}_{exp}^{(t,l)}$ is the actual potential decrease caused by hard reset. Note that $\mathbf{L}_{exp}^{(t,l)}$ is given by $[1 - \alpha(1 - \tau_{exp})]\mathbf{U}^{(t-1,l)}$ in Eq.(10), which means GLIF only performs hard reset on the exponential-decay-related part of the membrane potential. In a similar way, as $\gamma \to 0$, soft reset is incrementally biased, while $\gamma \to 1$, hard reset is incrementally biased. When $\gamma = 0$ or 1, GLIF supports the simplex reset formulation of spike initiation.

Therefore, with gating units, the GLIF-based spiking neurons can fuse the dual bio-features in each one of the three neuronal behaviors. When gating factors are binary, GLIF transforms into the corresponding simplex LIFs. For example, in the case of $\alpha, \beta, \gamma = 1, 0, 1$, GLIF is equivalent to the vanilla LIF. In a nutshell, GLIF unifies different simplex LIFs by incorporating dual bio-features and exploits gating factors to balance the dual bio-features.

**Channel-wise parametric method.** The parametric method is applied to all the membrane-related parameters in our GLIF, including primitives ($\tau_{lin}$, $\tau_{exp}$, $V_{re}$, $g^t$, and $V_{th}$) and gating factors ($\alpha$, $\beta$, and $\gamma$). Given $x$ is one of these membrane-related parameters, we can obtain $x$ by the Sigmoid function: $x = \frac{1}{1+\exp(-x_p)}$ where $x_p \in (-\infty, +\infty)$ is a learnable parameter that changes at each training iteration. With all these membrane-related parameters trainable, the adaptivity of the spiking neurons is increased. Such that, GLIF neurons can adjust their neuronal response characteristics in a wide range during training.

Furthermore, prior SNNs commonly adopt the same membrane-related parameters for all spiking neurons throughout the network, totally neglecting the heterogeneity. While in GLIF, the channel-wise sharing scheme, where spiking neurons in the same channel share a unique setting of the membrane-related parameters but inter-channel spiking neurons don't, is employed.

With the channel-wise sharing scheme and the parametric method combined, we can greatly increase the heterogeneity of spiking neurons after training. The heterogeneity over channels and layers can be observed as we visualize the membrane-related parameters in the Appendix.

**Training framework.** In terms of the training framework, we utilize backpropagation through time (BPTT) as the training algorithm for our GLIF SNNs. As for the non-differentiable Heaviside step

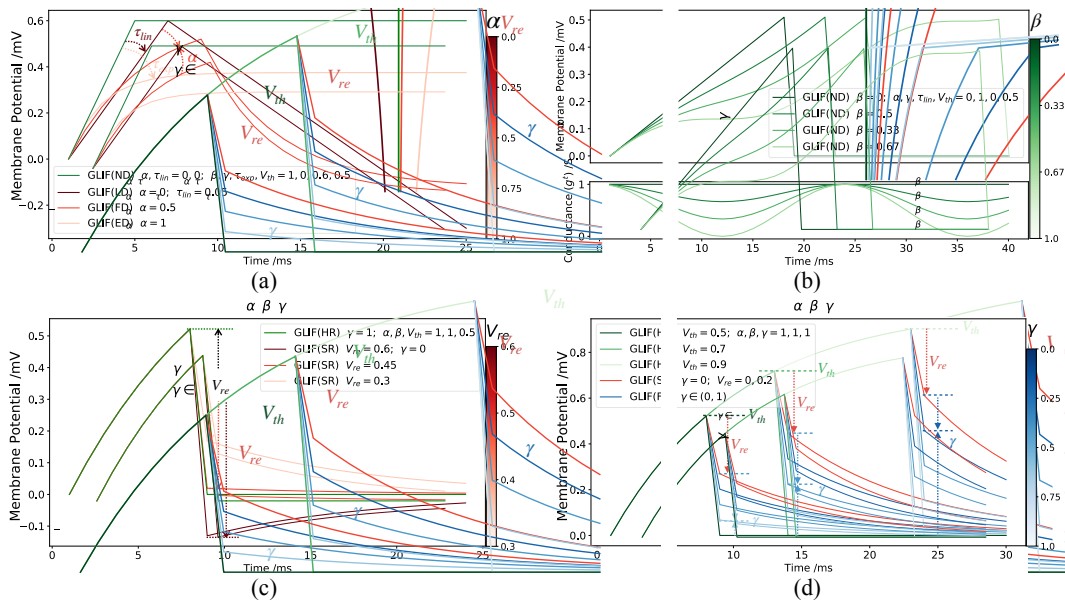

Figure 2: Visualization of neuronal dynamics. (a) The potential trace with $\alpha$ being the variable to bridge the linear and exponential decay. (b) **Upper:** the potential trace with $\beta$ being the variable to harness the amplitude of the cosine coding pattern. **Lower:** the cosine oscillation of conductance $g^t$. (c) The potential trace with $V_{re}$ being the variable to display possible scenarios of soft reset in comparison with the hard reset. (d) The potential trace with $\gamma$ being the variable to bridge the soft and hard reset. Note that **ND** represents non-decay, **LD** linear decay, **ED** exponential decay, **FD** fused decay, **SR** soft reset, **HR** hard reset, and **FR** fused reset.

function in Eq.(6), we solve it by the surrogate gradient defined as:

$$\frac{d\mathbb{H}(x)}{dx} = \mathbb{H}(0.5 - |x|).$$  (13)

Thus, GLIF SNNs can be trained end-to-end.

### 3.3   Dynamic Analysis

GLIF contains dual bio-features, which brings great diversity to the neuronal dynamics. Here we use a simple case to analyze the dynamic properties of GLIF. Fig. (2) illustrates the response of the spiking neurons to the constant input that spikes each time step until the membrane potential reaches the threshold. As shown in the figure, GLIF neurons with different parameter configurations have completely different dynamic responses to the same input.

**Bridging the gap between exponential decay and linear decay.**   As shown in Fig. 2(a), compared with non-decay, on the one hand, exponential decay brings the neuronal dynamic with stability and astringency, making it increasingly difficult for the membrane potential to climb over the threshold. However, the membrane potential falls into the saturation zone and fails to reach the potential threshold, which indicates exponential decay may impair the spiking responsiveness. On the other hand, linear decay operates equally at each time step to avoid saturation, thus maintaining the high spiking responsiveness. However, if there are no inputs, linear decay leads to the potential decreasing in an unlimited way, which impairs the stability. While in GLIF, the above two types of decay are the extreme cases, i.e., GLIF only acts exponential decay if $\alpha = 1$ and only serves linear decay if $\alpha = 0$. In most cases where $\alpha \in (0, 1)$, GLIF is eligible for exponential decay's astringency and benefits from the high spiking responsiveness linear decay brings about. To some extent, the gating factor $\alpha$ makes the two types of decay complementary and determines the proportion of each of the two decay, bridging the gap between the neuronal dynamics of exponential decay and linear decay and enlarging the representation space of membrane potential leakage. Furthermore, since $\tau_{exp}$ and $\tau_{lin}$ are learnable in GLIF, the gap between exponential and linear decay becomes more flexible. Therefore, the actual representation space given by $\tau_{exp}$, $\tau_{lin}$, and $\alpha$ could be even larger.

**Incorporating time-dependent synaptic weights.** In GLIF, we assign the time-dependent synaptic weights $g^t$ to the spike train to differentiate spike stimulation levels at different time steps. Since $g^t$ is learnable, the flexible-coding patterns of different GLIF neurons may differ significantly. For the convenience of analysis, we add constraints to $g^t$, assuming that $g^t$ is in the form of cosine oscillation, where $\beta$ controls the amplitude. As shown in Fig. 2(b), when $\beta = 0$, the uniform-coding pattern takes effect. Each time the spiking neuron receives an input spike, the membrane potential increases by the same amount. Our GLIF is different because the flexible-coding is also incorporated. When $\beta \in (0, 1)$, each time the GLIF receives an input spike, the membrane potential responds differently depending on the particular stimulation level of the current input spike. On the one hand, considering that there is no constraint on $g^t$ in practice, GLIF is more flexible, and the representation space is further enlarged. On the other hand, because the stimulus of input spikes at different times are considered differently, incorporating time-dependent synaptic weights may also help to better process temporal features.

**Bridging the gap between hard reset and soft reset.** Fig. 2(c) compares the neuronal dynamics between soft reset and hard reset. As we can see, soft reset reduces the current membrane potential by $V_{re}$. Since $V_{re}$ is not related to the current membrane potential, soft reset may lead to over-resetting or under-resetting of membrane potential, which is flexible for the spiking neurons. Hard reset reduces the current membrane potential to an empirical value, e.g., 0, restarting the potential trace all over, providing the neuronal dynamics with stability. While in GLIF, we consider the potential reduction caused by spike initiation to be both related to $V_{re}$ and the current membrane potential. Because $\gamma \in (0, 1)$ balances the amplitudes of the two resets, GLIF can reset the membrane potential to any possible levels between the consequent potentials of hard reset and soft reset as shown in Fig. 2(d), enlarging the representation space of spike initiation.

## 4 Experiments

In this section, we demonstrate the effectiveness of our proposed GLIF model. We first compare our results with other existing state-of-the-arts, then carry out ablation studies to evaluate different aspects of our proposed method.

### 4.1 Experimental Settings

We perform experiments on two kinds of datasets. The first is image classification datasets, including CIFAR [46] and ImageNet [47]. The other one is the event-based dataset, i.e., CIFAR10-DVS [48], which is converted from a popular image recognition dataset CIFAR-10 using a DVS camera. For fair comparisons with other methods, we choose the commonly used network architectures in existing works for experiments, including ResNet-family [49, 9, 44], 7B-wideNet [43], and CIFARNet [42, 8, 50], and apply tdBN [9]. More implementation details of training those SNNs are revealed in the Appendix.

### 4.2 Comparison with Existing Works

**CIFAR.** We test our GLIF on CIFAR-10 and CIFAR-100 datasets and run three times for each experiment to report the "mean $\pm$ std". As shown in Table 1, our GLIF SNNs can outperform existing works. Specifically, on CIFAR-10, we first test our GLIF on a smaller network CIFARNet and achieve the superior accuracy over other methods [42, 8, 50], even with fewer time steps. Since tdBN is employed in our GLIF SNNs, we then compare GLIF with STBP-tdBN [51] on ResNet-19. By simply changing the neurons in STBP-tdBN to GLIF, our SNNs can achieve significant improvement across all the tested time-steps. Compared with the state-of-the-art TET [12], our method can still obtain better accuracy, and our ResNet-19 with time-step 4 even outperforms theirs with time-step 6 by $0.35\%$. On CIFAR-100, our method can also achieve much better performance than STBP-tdBN and TET. We also compare GLIF ResNet-18 with the highly handcrafted ResNet-18 of Dspike [11]. As listed in Table 1, our GLIF ResNet-18 can outperform Dspike ResNet-18 by at least $0.63\%$ and $2.92\%$ absolute accuracy on CIFAR-10 and CIFAR-100, respectively.

**ImageNet.** We apply GLIF to ResNet-34 and MS-ResNet-18 [44]. Table 2 shows the results on the validation set. We first compare our method with STBP-tdBN, which can be viewed as our

Table 1: Comparisons with existing works on CIFAR datasets.

| Method | Architecture | CIFAR10 | | CIFAR100 | |
|---|---|---|---|---|---|
| | | TimeStep | Accuracy | TimeStep | Accuracy |
| STBP [42] | CIFARNet | 12 | 89.83 | - | - |
| STBP NeuNorm [8] | CIFARNet | 12 | 90.53 | - | - |
| TSSL-BP [50] | CIFARNet | 5 | 91.41 | - | - |
| STBP-tdBN [9] | ResNet-19 | 6 | 93.16 | 6 | 71.12±0.57 |
| | | 4 | 92.92 | 4 | 70.86±0.22 |
| | | 2 | 92.34 | 2 | 69.41±0.08 |
| TET [12] | ResNet-19 | 6 | 94.50±0.07 | 6 | 74.72±0.28 |
| | | 4 | 94.44±0.08 | 4 | 74.47±0.15 |
| | | 2 | 94.16±0.03 | 2 | 72.87±0.10 |
| Dspike [11] | ResNet-18* | 6 | 94.25±0.07 | 6 | 74.24±0.10 |
| | | 4 | 93.66±0.05 | 4 | 73.35±0.14 |
| | | 2 | 93.13±0.07 | 2 | 71.68±0.12 |
| | CIFARNet | 5 | 93.28 | - | - |
| GLIF | ResNet-19 | 6 | **95.03±0.08** | 6 | **77.35±0.07** |
| | | 4 | **94.85±0.07** | 4 | **77.05±0.14** |
| | | 2 | **94.44±0.10** | 2 | **75.48±0.08** |
| | ResNet-18 | 6 | **94.88±0.15** | 6 | **77.28±0.14** |
| | | 4 | **94.67±0.05** | 4 | **76.42±0.06** |
| | | 2 | **94.15±0.04** | 2 | **74.60±0.24** |

\* denotes a highly handcrafted network.

Table 2: Comparisons on ImageNet.

| Method | Architecture | T | Acc. |
|---|---|---|---|
| TET [12] | ResNet-34 | 6 | 64.79 |
| Dspike [11] | ResNet-34 | 6 | 68.19 |
| STBP-tdBN [9] | ResNet-34 | 6 | 63.72 |
| SEW+PLIF [43] | SEW-ResNet-34 | 4 | 67.04 |
| MS-ResNet [44] | MS-ResNet-18 | 6 | 63.10 |
| GLIF | ResNet-34 | 6 | **69.09** |
| | ResNet-34 | 4 | **67.52** |
| | MS-ResNet-18 | 6 | **68.11** |

**T** denotes the time step.

Table 3: Comparisons on CIFAR10-DVS.

| Method | Architecture | T | Acc. |
|---|---|---|---|
| STBP-tdBN [9] | ResNet-19 | 40 | 67.80 |
| LIAF [28] | LIAF-Net | 10 | 70.40 |
| LIAF+TA [33] | TA-SNN-Net | 10 | 72.00 |
| PLIF [10] | PLIF-Net | 20 | 74.80 |
| SEW+PLIF [43] | 7B-wideNet | 16 | 74.40 |
| GLIF | 7B-wideNet | 16 | **78.10** |

**T** denotes the time step.

baseline, on ResNet-34, and we can observe that our GLIF SNNs achieve better performance (69.09% v.s. 63.72%). Since parametric technology is employed in our GLIF, we also compare GLIF with the parametric spiking neuron PLIF [43]. Without changing the model structure, our method can still achieve higher accuracy than PLIF on ResNet-34 (67.52% v.s. 67.04%). Moreover, compared with MS-ResNet [44], whose handcrafted residual block achieves the state-of-the-art performance, we apply our GLIF to MS-ResNet-18 and can also significantly increase the accuracy (68.11% v.s. 63.10%). These results demonstrate the effectiveness of our proposed GLIF model. In addition, compared with other SNNs in TET [12] and Dspike [11], GLIF SNNs can also obtain higher accuracy.

**CIFAR10-DVS.** To demonstrate the effectiveness of GLIF in processing spatio-temporal features, we conduct experiments on the event-based dataset CIFAR10-DVS. Since SEW+PLIF [43] is the closest to our work for it also exploits parametric methods and tdBN, we use the same network architecture 7B-wideNet as SEW+PLIF. As shown in Table 3, GLIF outperforms SEW-PLIF along with other methods PLIF [10], STBP-tdBN [9], and LIAF [28], which demonstrates the effectiveness of the proposed GLIF on the event-based dataset.

Table 4: Comparisons with different simplex LIFs.

| Model | 000 | 001 | 010 | 011 | 100 | 101 | 110 | 111 | GLIF |
|---|---|---|---|---|---|---|---|---|---|
| ResNet-19 | 76.29 | 76.59 | 73.89 | 74.71 | 77.15 | 76.26 | 74.39 | 74.68 | **77.22** |
| ResNet-18 | 75.77 | 76.05 | 74.27 | 74.59 | 75.54 | 75.51 | 73.36 | 74.49 | **76.39** |

Table 5: Ablation study of gating factors.

| Model | T | GLIF_s | GLIF_f | GLIF |
|---|---|---|---|---|
| ResNet-19 | 4 | 76.89 | 71.26 | **77.22** |
| | 2 | 75.27 | 72.14 | **75.54** |
| ResNet-18 | 4 | 76.22 | 72.60 | **76.39** |
| | 2 | 74.24 | 72.10 | **74.85** |

**T** denotes the time step.

Table 6: Comparisons with Layer-wise GLIF.

| Model | T | Layer-Wise | GLIF |
|---|---|---|---|
| ResNet-19 | 4 | 75.71 | **77.22** |
| | 2 | 74.19 | **75.54** |
| ResNet-18 | 4 | 75.29 | **76.39** |
| | 2 | 73.40 | **74.85** |

**T** denotes the time step.

### 4.3 Ablation Study

**Effectiveness of different bio-features.** To demonstrate the effectiveness of different bio-features that are incorporated in our GLIF, we evaluate the performance of GLIF SNNs in extreme cases where GLIF only acts single bio-feature in each neuronal behavior. Since the gating factors control bio-features, we can obtain eight simplex LIFs by combining different gated biometrics, all of which have static binary gating factors. We name those simplex LIFs with the values of their gating factors, e.g., 101 is the GLIF with $\alpha, \beta, \gamma = 1, 0, 1$ that only acts exponential decay, the uniform-coding pattern, and hard reset. On CIFAR-100, we use ResNet-18 and ResNet-19 with time-step 4 for experiments. As listed in Table 4, our proposed GLIF outperforms all of those simplex LIFs, which demonstrates the effectiveness of fusing different bio-features.

**Importance of learnable gating factors.** To demonstrate the effectiveness of our proposed learnable gating factors in fusing bio-features, we compare GLIF with other two GLIF variants: GLIF_s and GLIF_f, where GLIF_s keeps gating factors static after initialization, GLIF_f removes gating-factors and simply fuses two different bio-features by addition. On CIFAR-100, we use ResNet-18 and ResNet-19 with time-step 2 and 4. Table 5 lists the experimental results. On the one hand, compared with GLIF_s, our method can get higher accuracy, which demonstrates that learnable gating factors can make GLIF better. On the other hand, compared with GLIF_f, GLIF can also achieve better performance, this is because learnable gating factors can balance dual bio-features in each neuronal behavior. As a result, learnable gating factors are important in GLIF.

**Channel-wise v.s. Layer-wise.** Besides the investigations above, we conduct an ablation study on GLIF with different parameter-sharing schemes, which are layer-wise sharing and channel-wise sharing. We also use ResNet-18 and ResNet-19 with time-step 2 and 4 on CIFAR-100. As listed in Table 6, GLIF, which leverages channel-wise sharing scheme, consistently improves the performance over the layer-wise sharing scheme.

## 5 Conclusion

In this paper, we propose GLIF, a unified spiking neuron, that fuses the bio-features in each neuronal behavior. In GLIF, our method exploits the gating factor to bridge the gap between the dual bio-features, enlarging the representation space. Furthermore, we adopt the channel-wise parametric method to make all membrane-related parameters learnable, and train those parameters through BPTT, increasing the neuronal adaptivity of spiking neurons and enhancing the neuronal dynamic diversity of spiking neurons. We validate our GLIF SNNs on various benchmark datasets, showing the state-of-the-art performance compared with other methods. Therefore, we think GLIF could be a positive answer to the issue: *could the higher neuronal dynamic diversity of spiking neurons help SNNs do better?*

Additionally, the distributions of learned parameters are interesting as we visualize them in the Appendix. The initially identical parameters learn into different bell-shaped distributions layer-wisely. We think this could be further studied to mine the biological plausibility of SNNs. How to apply GLIF in other training frameworks, e.g., ANN2SNN conversion, also remains a worth-studying problem.

## Acknowledgments and Disclosure of Funding

This work was supported in part by the National Key Research and Development Program of China (Grant No. 2021ZD0201504), the Strategic Priority Research Program of Chinese Academy of Science (No.XDB32050200), Jiangsu Key Research and Development Plan (No.BE2021012-2).

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
