# A Appendix

## A.1 Visualization of Learned Parameters

This section visualizes the distributions of the learned parameters from ResNet-18 on CIFAR-100.

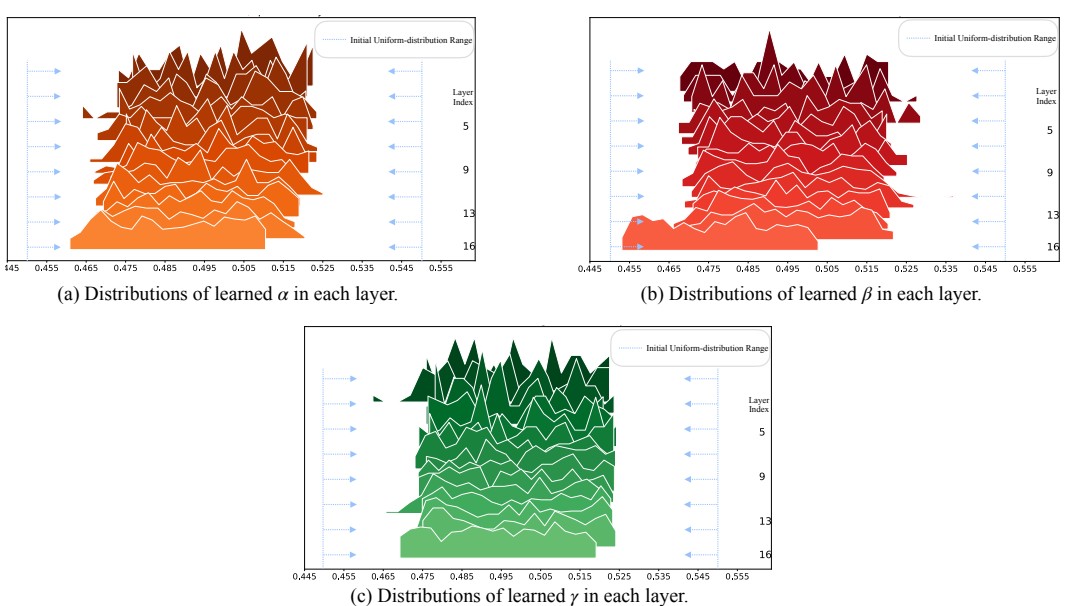

Figure 1: Distributions of learned gating factors over channels in each layer.

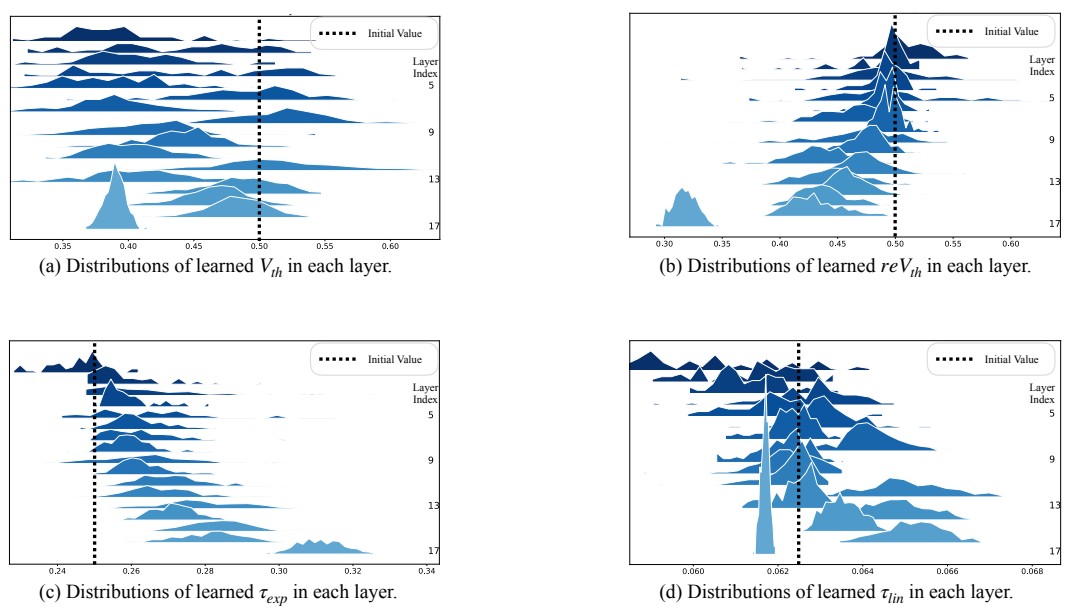

Figure 2: Distributions of learned primitives over channels in each layer.

Fig. 1 illustrates the distributions of learned gating factors $\alpha$, $\beta$, and $\gamma$. Initially, they are all uniformly sampled from [0.4502, 0.5498). After training, the property of uniform distributions remains, but the ranges are all narrowed and the displacement of each gating factor is rather small. This indicates dual bio-features of each behavior tend to be considered equally effective for the whole SNN. Such that, compared to the single bio-feature, the hybrid bio-feature are inclined to by the SNN.

Fig. 2 illustrates the distributions of learned primitives $V_{th}$, $reV_{th}$, $\tau_{exp}$ and $\tau_{lin}$. Two interesting phenomena can be observed. For one thing, neurons in the same layer are inclined to the same displacement direction and the distribution of their primitives naturally form into the phenotypes of bell-shaped distributions. This is quite similar to the hierarchical structures of brains. For another, as a whole, $V_{th}$ tends to be smaller which indicates the potential required for a spike is smaller. $reV_{th}$ tends to be smaller which indicates the potential loss for a soft-reset is smaller. $\tau_{exp}$ tends to be bigger which indicates the potential loss for decay is smaller. $\tau_{lin}$ overall tends to remain still. Overall, the potential loss becomes smaller and the potential required for a spike becomes less. Therefore, the whole SNN learns to be more active or excitatory, which is similar to the on-task state of brains.

Above all, the heterogeneity of the GLIF-based ResNet-18 is proven greatly increased through visualizing the distributions of the learned parameters. Plus, the learned parameters somehow embodies some biological properties of brains. Such observations could be further studied to mine the biological plausibility of SNNs.

## A.2  Implementation Details

**Training details.** For all of our experiments, we use the stochastic gradient descent (SGD) optimizer with a momentum of $0.9$ and weight decay of $5e-5$ to optimize the model. No weight decay is applied on the parameters in GLIFs. The learning rates of the GLIF parameters and the model parameters are both reduced by the cosine-annealing scheduler. Specially for CIFAR, gating factors' learning rate is set to $1/10$ of other parameters'. Specially for CIFAR10-DVS, no weight decay is applied to the whole SNN and the annealing period ($T_{max}$) of the cosine-annealing scheduler is 64. As for the initialization of GLIF parameters, gating factors are initialized with the random values sampled from $[0.4502, 0.5498)$, while the primitives are initialized as listed in Table 1. We train all of the models on A100 GPUs, and other details for training different models are listed in Table 2.

Table 1: Initial settings for GLIF parameters.

| Parameter | $V_{th}$ | $V_{re}$ | $g^t$ | $\tau_{exp}$ | $\tau_{lin}$ |
|:---:|:---:|:---:|:---:|:---:|:---:|
| **CIFAR** | 0.5 | 0.5 | 0.5 | 0.25 | 0.0625 |
| **ImageNet** | 0.5 | 0.5 | 0.5 / 0.9 | 0.25 | 0.0625 |
| **CIFAR10-DVS** | 1.0 | 1.0 | 0.9 | 0.5 | 0.03125 |

For ImageNet, $g^t$ is set 0.5 for time-step 4, and 0.9 for time-step 6.

Table 2: Training hyper-parameters for GLIF SNNs.

| Architecture | Initial Learning Rate | Total Epoch | Batch Size per GPUs | GPUs |
|:---:|:---:|:---:|:---:|:---:|
| **CIFARNet** | 0.1 | 200 | 64 | 1 |
| **ResNet-18** | 0.1 | 200 | 64 | 1 |
| **ResNet-19** | 0.1 | 200 | 64 | 1 |
| **ResNet-34** | 0.1 | 150 | 50 | 4 |
| **MS-ResNet-18** | 0.1 | 150 | 50 | 4 |
| **7B-wideNet** | 0.01 | 200 | 32 | 1 |

**Architecture.** ResNet-18 is originally designed for ImageNet task in [1]. To process CIFAR tasks, we remove the max-pooling layer, replace the first $7 \times 7$ convolution layer with a $3 \times 3$ convolution layer, and replace the first and the second 2-stride convolution operations as 1-stride, following the modification philosophy from STBP-tdBN [2].

7B-wideNet [3] is originally designed to receive frames with $128 \times 128$ pixels in each channel (two in total). Whereas, we resize each frame into $48 \times 48$ pixels in our DVS data pre-processing. Therefore, we reschedule the down-sampling strategy so that 7B-wideNet is constructed as *c64k3s1-BN-GLIF-{SEW Block (c64)}\*4-APk2s2-c128k3s1-BN-GLIF-{SEW Block (c128)-APk2s2}\*3-FC11*, where c32k3s1 represents the convolutional layer with kernel size 3 stride 1, APk2s2 is the average pooling with kernel size 2 and stride 2, SEW Block is proposed and illustrated in [3], and the symbol {}\* $n$ denotes the structure {} is repeated $n$ times. Notably, our implemented 7B-wideNet is the same size of [3], but has fewer operations due to the smaller input size.

Except for the above networks, we use the same architectures as the prior works.

**Data Pre-processing for CIFAR10-DVS.** We first separate the whole dataset into 9000 training samples and 1000 test samples. In the data pre-processing of single sample of CIFAR10-DVS, we split the event stream of the sample into 16 slices and integrate the events in each slice into one frame, and resize each frame into $48 \times 48$ pixels.

### A.3 Coarsely Fused LIF

In this section, we formulate the GLIF_f, which is a **coarsely fused LIF** without gating factors and utilized as a comparison in the ablation study. In the GLIF_f, primitives are directly stacked together as follows:

$$\mathbf{U}^{(t,l)} = \mathbf{L}^{(t,l)} + \mathbf{I}^{(t,l)} + \mathbf{F}^{(t,l)} \odot \mathbf{S}^{(t-1,l)}, \tag{1}$$

$$\mathbf{C}^{(t,l)} = \mathbf{W} \cdot \mathbf{S}^{(t,l-1)}, \tag{2}$$

$$\mathbf{S}^{(t,l)} = \mathbb{H}(\mathbf{U}^{(t,l)} - V_{th}), \tag{3}$$

$$\mathbf{L}^{(t,l)} = \tau_{exp} \mathbf{U}^{(t-1,l)} - \tau_{lin}, \tag{4}$$

$$\mathbf{I}^{(t,l)} = g^t \mathbf{C}^{(t,l)}, \tag{5}$$

$$\mathbf{F}^{(t,l)} = -\tau_{exp} \mathbf{U}^{(t-1,l)} - V_{re}. \tag{6}$$

Although, the representation space of GLIF_f is larger than GLIF due to the removal of the gating mechanism. As shown in the ablation study, GLIF_f performs even worse than GLIF_s. This indicates that the balance between dual primitives is essential in regulating the representation space within a valid range.