# OpenReview forum: "GLIF: A Unified Gated Leaky Integrate-and-Fire Neuron for Spiking Neural Networks"
_NeurIPS.cc/2022/Conference — NeurIPS 2022 Accept_

### Official Review · Reviewer_i4xG · 2022-07-05

**Rating:** 5
**Confidence:** 4
**Soundness:** 3 good
**Presentation:** 3 good
**Contribution:** 3 good

**Summary:**

In this work, the authors propose a mixture type of neuron model called GLIF to introduce additional flexibility to the neurons. They demonstrate that the proposed model can achieve SOTA performances on a bunch of classification tasks.

**Questions:**

1. How does the distribution of $\alpha$, $\beta$, and $\gamma$ look like? Does it display heterogeneity over layers, especially for the phenotypes defined in Figure 2?
2. Since each neuron here contains three branches, it makes me feel that its computational cost is actually equivalent to three times the regular one, where the advantage of the current approach is not as high as it appears to be.
3. Considering the model complexity, it would also be interesting to compare with the model ensembled with three different configurations for a single neuron type in order to support the claim that the heterogeneity in the same model makes a difference.

**Strengths And Weaknesses:**

Strengths:
1. The paper is written clearly.
2. The model achieved SOTA performances.
Weakness:
1. There could be more analyses on why the proposed method works beyond the comparison to other methods.

---

> ### Author Response · Authors · 2022-08-02
> **Response to Reviewer i4xG**
>
> Thank you for your insightful feedback.  We list your questions and give our detailed answers.
>
> >  Con. 1: There could be more analyses on why the proposed method works beyond the comparison to other methods.
>
> A. 1: Most of our main competitors are those works that propose effective learning methods to train SNNs to reach a high performance.  The comparable work as far as we know is PLIF, which adopts the parametric method.  The superiority of GLIF, compared to PLIF, mainly lies in the incorporation of different bio-features and the further exploitation of the parametric method.
>
> > Q. 1: How does the distributions of α, β, and γ look like? Does it display heterogeneity over layers, especially for the phenotypes defined in Figure 2?
>
> A. 2: We visualize the distribution of gating factors along with other learnable parameters, and add the visualized figures and the corresponding analysis into the supplementary material.  The heterogeneity can be observed through the distributions over layers.  Two interesting observations can also be found. For one thing, neurons in the same layer are inclined to the same displacement direction and the distribution of their primitives naturally form into the phenotypes of bell-shaped distributions. This is quite similar to the hierarchical structures of brains. For another, the whole SNN learns to be more active or excitatory, which is similar to the on-task state of brains. We think this could be further studied to mine the biological plausibility of SNNs.
>
> Fig. 2 mainly illustrates the dynamic properties of GLIF on a macro level with hundreds of simulation time steps .  Since our simulation tilme step is very short in actual SNNs, the learned dynamic characteristics need to fit such a short simulation time steps and be helpful to improve the SNN as a whole.  Therefore, there inevitably exists a gap between the learned phenotypes and that defined in Fig. 2.
>
> > Q. 2: Since each neuron here contains three branches, it makes me feel that its computational cost is actually equivalent to three times the regular one, where the advantage of the current approach is not as high as it appears to be.
>
> A. 3: You share the same concern with reviewer ZaJt.  We address your common concern in General Response.
>
> > Q. 3: Considering the model complexity, it would also be interesting to compare with the model ensembled with three different configurations for a single neuron type in order to support the claim that the heterogeneity in the same model makes a difference.
>
> A. 4:  Each neuronal behavior supports two different biological features, and there are three neuronal behaviors in total. Therefore, there exists eight different configurations for a single neuron type (referred to as eight simplex LIFs).  In Table 4, ablation study. We compare these eight different simplex LIFs with GLIF. The best simplex LIF varies from ResNet-18 to ResNet-19. This implies the simplex LIFs may have compatibility issues with different model architectures.While GLIF achieves superior performance to all the simplex LIFS as GLIF covers all their representation spaces.

---

> > ### Comment · Reviewer_i4xG · 2022-08-08
> > **Further comments**
> >
> > Thanks for the response and clarification.
> >
> > In terms of Q3, If I understand it correctly, each neuron in GLIF takes different $\alpha$, $\beta$ and $\gamma$. Thus, to support the claim of heterogeneity, it would be nice to have all neurons take the same tuple $(\alpha_0, \beta_0, \gamma_0)$ as an alternative. As mentioned by other reviews, it is important to verify which part contributes to the improvement.

---

> > > ### Author Response · Authors · 2022-08-09
> > > **Further Response to Reviewer i4xG.**
> > >
> > > Thanks again for your very helpful review of this work. We hope the following response and additional ablation study would address your concerns.
> > >
> > > 1. Yes, each neuron in GLIF takes different $(\alpha,\beta,\gamma)$ to increase the heterogeneity.
> > >
> > > 2. In Table 5, we verify the contributions of learnable primitives and gating factors. In GLIF_f, we remove the gating factors and keep primitives learnable. The result shows GLIF far outperforms GLIF_f, which demonstrates the importance of the gating mechanism.
> > >
> > > 3. To further verify which part of the gating mechanism contributes more to the improvement (different values or learnable values), we conduct the following experiments on CIFAR-100 with TimeStep 4.
> > >
> > > |Model|Same&Static Gating Factors| Uniformly Sampled&Static Gating Factor|GLIF|
> > > |:---:|:---:|:---:|:---:|
> > > |ResNet-18 |76.22|76.03|**76.39**|
> > > |ResNet-19|76.85|76.89|**77.22**|
> > >
> > >  **Same&Static Gating Factors** (referred to as *S.S.* for short) means we assign $(0.5, 0.5, 0.5)$ to $(\alpha,\beta,\gamma)$ and freeze them during training. **Uniformly Sampled&Static Gating Factors** (referred to as *U.S.* for short) means we assign uniformly random values to $(\alpha,\beta,\gamma)$ and freeze them during training. While, in GLIF, $(\alpha,\beta,\gamma)$ are uniformly random at the initialization phase but learnable during training. Comparing  *S.S.* and *U.S* indicates that randomly increasing the heterogeneity doesn't help improve SNNs. Making the gating factors learnable is important.

---

### Official Review · Reviewer_jY7N · 2022-07-10

**Rating:** 5
**Confidence:** 5
**Soundness:** 3 good
**Presentation:** 3 good
**Contribution:** 3 good

**Summary:**

This paper proposes the GLIF neuron model, which has rich neuronal dynamics with learnable gates. The GLIF neuron achieves SOTA accuracy on CIFAR10, CIFAR100 and ImageNet datasets.

**Questions:**

Does g^t in line 153 represent G_beta in eq.11? Or it is a learnable parameter?
Why beta=1 in eq.11 means single spike coding pattern of integration accumulation? In my opinion, when beta=1, G_beta = g^t * C, which behaves like (1 – tau) * C in the LIF neuron (V = (1-tau) * V + tau * C).

In line 167, “GLIF only performs hard reset on the exponential-decay-related part of the membrane potential”. Can the author explain why use such a hard reset? In most cases, hard reset means the whole V, rather than a part of V, is set to V_reset .

**Limitations:**

I find from codes that some parameters of GLIF are not reused across different time-steps because they have shape [T, *] where T is the total time-steps. I am afraid that this design violates the parameter reuse rule in SNNs.

**Strengths And Weaknesses:**

The complex spiking neuron model has not been well studied in deep SNNs. Most deep SNNs use the simple IF or LIF neuron. As far as I know, this is one of the first papers that focuses on the complex spiking neuron.

The design of the GLIF neuron is reasonable. And it gets SOTA accuracy on most datasets.

I notice that the proposed method gets 100% accuracy on DVS Gesture, which is hard to achieve. I check the codes and find that there is a bug at line 241 in data_builder.py:

test_dataset = DVS128Gesture(dataset_dir, train=True

It seems that the author takes train set as the test set and gets 100% accuracy. But I think it is an honest mistake.

---

> ### Author Response · Authors · 2022-08-02
> **Response to Reviewer jY7N**
>
>
> Thank you for your very careful feedback.  We list your questions and give our detailed answers.
>
> > Con. 1: Bug issue
>
> A. 1: We have to admit that we committed a very rookie but severe mistake in coding the DVS Gesture dataset builder.  Consequently, the DVS Gesture experiments in our paper is invalid. We would like to honestly address this mistake.  So, we post the general response to explain how we mend this up. Please see our general response.
>
> > Q. 1:  Does g^t in line 153 represent G_beta in eq.11? Or it is a learnable parameter? Why beta=1 in eq.11 means single spike coding pattern of integration accumulation? In my opinion, when beta=1, G_beta = g^t * C, which behaves like (1 – tau) * C in the LIF neuron (V = (1-tau) * V + tau * C).
>
> A. 2: $g^t$ denotes the conductance primitive, which functions as the time-wise weights assigned to each time step.  $\mathbb{G_{\beta}}$ is the gating unit to compute the incremental potential caused by integration accumulation.  $g^t$ is a learnable parameter, whose value varies at each time step $t$ .  So, actually $g^t$ represents the weight vector on the time dimension.
>
> When $\beta =1$, Eq.(11) becomes $\mathbb{G_{\beta}}() = g^t C^{(t,l)}$.  When $\beta =0$, Eq.(11) becomes $\mathbb{G_{\beta}}() = C^{(t,l)}$.  In the former case, the learned coding scheme is fully accepted, while in the latter one, spike input are considered uniformly equal along the time dimension. Therefore, we call the former case the flexible-coding scheme as the actual values of  $g^t$ could be flexible, and the latter one the uniform-coding scheme. By "single spike coding pattern", we mean when $\beta$ is binary, only one of the aforementioned coding scheme is adopted.
>
> To better describe the two different neuronal behaviors, we decouple the exponential decay primitive $\tau$ from the integration accumulation. Such that, LIF Neuron ($V = (1-\tau) * V + \tau*C$) is re-formulated as $V=\tau * V + C$.  With $g^t$ added, it becomes $V^t =\tau  V^t + g^t C$ .
>
> > Q. 2: Can the author explain why use such a hard reset? In most cases, hard reset means the whole V, rather than a part of V, is set to V_reset .
>
> A. 3: As LIF models usually do not consider the refractory period, which is also a biological feature, we hope to introduce such a bio-feature in an implicit way.  So, we view the refractory period as prior knowledge and bring it into the modeling of the GLIF reset behavior.
>
> Assuming hard reset is conducted on the whole part of membrane potential, Eq. (12) becomes $\mathbb{G_{\gamma}}(\mathbf{L}^{(t,l)}; V_{re}, \mathit{0})   = -(\mathit{0}+\gamma) \mathbf{L}^{(t,l)}- (1-\gamma)V_{re}$, where $\mathbf{L}^{(t,l)} = [1-\alpha(1-\tau_{exp})] \mathbf{U}^{(t-1,l)} - (1-\alpha)\tau_{lin}$.  Obviously, $\mathbf{L}^{(t,l)} < \mathbf{L}^{(t,l)}_{exp} $.
>
> Such that $\mathbb{G_{\gamma}}(\mathbf{L}^{(t,l)};.)$ > $\mathbb{G_{\gamma}}$ ( $\mathbf{L}^{(t,l)}_{exp} $;.).
>
>  And $\mathbb{G_{\gamma}}$ ( $\mathbf{L}^{(t,l)}_{exp} $;.)  is adopted in GLIF.  As a result, GLIF always tends to reset more potential, which makes potential required for the next spike is a bit more.  Such being the case, the firing of the next spike could be possibly delayed, which matches the philosophy of the refractory period.
>
> > Q. 4: I find from codes that some parameters of GLIF are not reused across different time-steps because they have shape [T, *] where T is the total time-steps. I am afraid that this design violates the parameter reuse rule in SNNs.
>
> A. 5: This is the only parameter that is not reused across different time-steps, which is $g^t$. This parameter functions as time-wise convolution. During each inference across the time dimension, it only operates weight assignment to the input of the current time-step, and doesn't access temporal inputs ahead of time. Therefore, we are convinced that such mechanism remains biologically plausible. We also find a similiar mechanism has already been applied in a prior work [1]. Their proposed temporal-wise attention module is not reused across time as well.
>
>
> [1] Yao M, Gao H, Zhao G, et al. Temporal-wise attention spiking neural networks for event streams classification[C]//Proceedings of the IEEE/CVF International Conference on Computer Vision. 2021: 10221-10230.

---

### Official Review · Reviewer_GThD · 2022-07-11

**Rating:** 3
**Confidence:** 5
**Soundness:** 2 fair
**Presentation:** 2 fair
**Contribution:** 2 fair

**Summary:**

The authors proposed three attention schemes that work on three different aspects of the LIF model: membrane potential leakage, integration accumulation, and membrane potential rest. The authors aim to enhance the LIF model's representation space with the proposed attention schemes. The key to the proposed attention schemes is three learnable parameters. They balance the pre-defined LIF behaviors, i.e., linear decay vs. exponential decay, uniform-coding vs. flexible-coding, and hard reset vs. soft reset rest.

To show the effectiveness of the proposed method, the authors mainly leveraged ResNet-based architectures on classification tasks.

**Questions:**

writing issues:
1. line 41: "units $G_\alpha$, $G_\beta$, and $G_\gamma$ for those three neuronal behaviors" It is unclear to me which one is which.
2. lines 58-59. Poor writing


**Limitations:**

From my perspective, the proposed method is incremental, not exciting. More importantly, the learnable parameters did not really solve the "single biological feature" problem. The proposed method added three dimensions to the LIF model but is still a static SNN model.

The validation is weak from my perspective. Comparing with SRM is necessary, and the validation without tbBN is necessary. The experimental results related to the spiking threshold should also be reported and analyzed.

**Strengths And Weaknesses:**

Strengths:
+1. Enhancing SNNs' capability in encoding different bio-inspired behavior is an exciting direction.

Weaknesses:
-1. I did not see the value of adding many learnable parameters into LIF. Essentially, they are still fixed during inferences and "support only a single biological feature in different neuronal behaviors." In addition, the SRM model has already offered much more parameters, which can be learned directly. As such, the authors should provide a comparison with SRM-based models.

-2. I think the key to SNNs research should be on the dynamic side. Dynamically dealing with different conditions is more appreciated. However, the proposed method did not introduce any dynamic schemes.

-3. The authors applied tdBN to all their models. Did other competing models have tdBN? In addition, what's the performance without tbBN. Basically, we do not know whether the tbBN gives more contribution to the performance or the proposed method.

-4. How does the spiking threshold impact the classification performance? In theory, all the proposed attention schemes can be replaced by applying different spiking threshold to each SNN neuron. Therefore, it is necessary to show the experimental results related to different spiking thresholds.

---

> ### Author Response · Authors · 2022-08-02
> **Response to Reviewer GThD**
>
>
> Thank you for your very critical feedback. We hope our detailed reponse can further explain our methods, and address your concerns about our work.
> > Con. 1:  I did not see the value of adding many learnable parameters into LIF.
>
> A. 1: The idea of adding more parameters is meant to incorporate different biological features. The idea of making all these membrane-related parameters learnable, which is realized by the channel-wise parametric method, is meant to leverage the larger representation space of GLIF to increase the neuronal dynamic diversity of spiking neurons. **From the perspective of the whole SNN**, the hybrid bio-features should be quite different among the GLIF-based neurons. Therefore, the supported biological features are in proportion to the SNN size. This is also the reason why deep SNNs are appreciated in this work.
>
> The hybrid rule embodies the philosophies of different biological features. Taking the example of reset.  Hard reset is more adaptive, only considering the current membrane potential.  While soft reset is more empirical, only considering a pre-set prior value.  And, GLIF considers both.
> > Con. 2:  In addition, the SRM model has already offered much more parameters, which can be learned directly.
>
> A. 2:  As far as we know, in the frameworks of neural networks, $SRM_0$ model is commonly adopted and can be viewed as an alternative to the vanilla LIF.  The formulation of $SRM_0$ are as follows: [1] [2]
>
> $a^{(l)}(t) = (\epsilon_d * s^{(l)})(t)$
>
> $u^{(l+1)}(t) = W^{(l)} a^{(l)}(t) + (v * s^{(l+1)})(t)$
>
> $s^{(l+1)} = f_s(u^{(l+1)}(t))$
>
> The only difference is the extra kernel functions $\epsilon_d$ and $v$. Practically,  these kernel functions usually has only one extra parameter is added to each function [4]. Therefore, with only two more parameters and two more temporal convolutional operations added, $SRM_0$ is quite computationally costly and does not offer much more parameters as expected.
>
>  Furthermore, we also conduct additional experiments  to compare GLIF and $SRM_0$ :
> |Dataset | Model |  GLIF (T)   | TSSL-BP (T) [3] | STiDi-BP (T)  [4] |
> |  ---- | ----  | ----  |----  |----  |
> |CIFAR-10|CIFARNet (W/o BN)|  91.62 (5)   | 91.41 (5)  |---|
> |MNIST|784-400-10 FC|  97.74 (50)   |---|97.4 (100) |
> >  Con. 3: I think the key to SNNs research should be on the dynamic side.
>
> A. 3: Our motivation is to figure out the issue: "could higher neuronal dynamic diversity of spiking neurons help SNNs do better?"  We propose GLIF to enlarge the representation space. With the channel-wise parametric method, the neuronal dynamic diversity of GLIF-based SNNs is increased.  Even with very naive training algorithm, our GLIF-based SNNs can achieve superior performance. Dynamic method is a important research direction, but research on other directions also counts for SNN.
> >  Con. 4: The authors applied tdBN to all their models. Did other competing models have tdBN? In addition, what's the performance without tbBN.
>
> A. 4: Our main competitors are STBP-tdBN, TET, DSpike in Table 1. Our main competitors are STBP-tdBN, SEW + PLIF, TET, Dspike, and MS-ResNet in Table 2 on ImageNet benchmark.  **All of them adopt tdBN.** So we believe our experimental settings are fair enough to demonstrate the effectiveness of the proposed GLIF.
> We also conduct the additional ablation study on the contribution of tdBN.  We use CIFARnet on CIFAR-10 benchmark:
> | / | Vanilla LIF (T=5)| GLIF (T=5) |
> |  ----  | ----  |----  |
> |w/o BN|90.63|91.62|
> |w/ BN|90.46|92.42|
> |w/ tdBN|91.43|93.82|
>
> Without tdBN, our GLIF-based CIFARNET still achieve the state-of-the-art. And our GLIF can effectively benefit from BN techniques.
> > Con. 5: How does the spiking threshold impact the classification performance? In theory, all the proposed attention schemes can be replaced by applying different spiking threshold to each SNN neuron.
>
> A. 5: Actually, our spiking threshold is also a learnable parameter. As the gating factors only control the balance between the fused bio-features rather than the amplitude of the neuronal behavior, the relationship between the attention schemes and the spiking threshold is uncertain, and the theoretical replacement seems unfeasible.  Our ablation study in Table 5, where GLIF_f removes all the gating factors, GLIF_s adds the static gating factors to GLIF_f. The results shows, the performance of GLIF > GLIF_s > GLIF_f, which demonstrates the effectiveness of the proposed attention schemes.
>
> > Writing issues
>
> A. 6: We fix these issues in the revivsed paper.
>
> [1] S.M. Bohte, J.N. Kok, H. La Poutre, Error-backpropagation in temporally encoded networks of spiking neurons
>
> [2] Shrestha S B, Orchard G. Slayer: Spike layer error reassignment in time
>
> [3] Zhang W, Li P. Temporal spike sequence learning via backpropagation for deep spiking neural networks
>
> [4] Mirsadeghi M, Shalchian M, Kheradpisheh S R, et al. STiDi-BP: Spike time displacement based error backpropagation in multilayer

---

> ### Author Response · Authors · 2022-08-05
> **Response to Reviewer GThD. Further clarification on our work.**
>
> Thanks for your insightful feedback and your time in reading our paper. Here we want to clarify more on our work to further address your concerns. And, we really hope that you would re-consider your rating.
>
> 1. Our motivation is to figure out the issue of "**could the higher neuronal dynamic diversity of spiking neurons help SNNs do better?**". The model GLIF that we propose incorporates dual potential changing rules in terms of each biological behavior. Through training, each GLIF-based neuron can learn to decide "which rule is inclined, and to which degree the amplitude of each rule should be". Such that, after training, different GLIF-based neurons are completely different on response characteristics, and the neuronal dynamic diversity in SNNs is greatly increased.  That is why so many learnable parameters are needed.
>
> 2. It is quite obvious that deep SNNs show much more potential for higher neuronal dynamic diversity because of more layers, channels and more different neurons. So, in our work, **deep SNNs are appreciated**. While, $SRM$, compared to LIF-based models, is not computational friendly due to temporal convolution operations, does not add much more learnable parameters as expected, and does not prove to have better compatiblity with deep SNNs referring to prior works. Such that, SRM-based models are not considered in our work.
>
> 3. Dynamic methods are **not** our focus.  Yes, it is a very important research direction for SNNs, but it is not the only important one. We are convinced that the research directions for SNNs should be diverse. Besides, with more parameters added, GLIF naturally provides a breeding ground to research more dynamic or adaptive methods on these paremeters.
>
> 4. tdBN is a very practical technique to make SNNs capable of going deeper. This technique has been frequently utilized in recent years. **All our competitors in comparisons adopt such technique.** Plus, our competitors mainly exploit sophisticated learning methods to improve SNNs, while we only use the very naive surrogate gradient substitution. Such that, we believe our experiments are fair enough to compare with existing works.
>
> 5. As far as we know, **no rigorous theory could support the equivelent replacement** of gating factors by different thresholds. The spiking threshold alone, can hardly decide any properties of a spiking neuron. With all other primitives combined, we can roughly say a spiking neruon is excitatory or inhibitory. Besides, the spiking threshold is also a learnable parameter in GLIF, the actual learned values of thresholds are quite different as we visualize them in the Supplementary Material.  Above all, replacing the gating mechanism by applying different spiking threshold seems not feasible.

---

> ### Comment · Reviewer_GThD · 2022-08-09
> **Thanks for your responds**
>
> Thanks for your responses and efforts.
>
> However, the responses do not address my concerns well.
>
> 1. "Essentially, they are still fixed during inferences and 'support only a single biological feature in different neuronal behaviors.' " still hold.
>
> 2. learnable threshold is still a fixed threshold, different from the dynamic threshold. Yes, it solved the concern about the impacts of the threshold. I agree with the authors that  "no rigorous theory could support the equivalent replacement." However, it does not mean it would not work. The proposed work is experimental research essentially, without rigorous theory support either.
>
> 3. LIF is a bio-inspired model; adding many learnable parameters lacks justification. I disagree with the authors that SRM is an alternative to the vanilla LIF. Essentially, LIF is a special case of SRM.
>
> 4. "Dynamic methods are not our focus. Yes, it is a very important research direction for SNNs, but it is not the only important one." Of course! However, adding many learnable parameters is not an important one from my perspective. More importantly, it lacks justification. Then, why not we add more parameters? How many more is the best?

---

> > ### Author Response · Authors · 2022-08-09
> > **Response to your further comments.**
> >
> > Thanks for your time in commenting and further clarifying your concerns. We want to add more responses to your concern. And, we realy really hope that you would re-consider your rating.
> >
> > 1. From the perspective of a single spiking neuron, it is true that a GLIF-based neuron only behaves with fixed bio-features. But, from the perspective of the whole SNN, GLIF-based SNNs have much more increased heterogeneity compared to other LIF-based SNNs. How to adaptively or dynamically choose different bio-features for a single neuron during inference could be a very prospective research direction, but we mainly focus on the impact of the increased heterogeneity in this work. To this end, we need to add extra parameters and make them different.
> >
> > 2. Dynamic threshold could be very effective in improving SNNs. But as stated above, the dynamic method is not our focus, and it is hard to make a fair comparison between GLIF and other methods with dynamic thresholds. Our earlier response to this is to address the concern of lacking comparisons with dynamic thresholds.
> >
> > 3. We did not add parameters by random. We reasonably add one extra parameter **along the three critical neuronal behaviors**. Such that, the dual bio-features can be formed and the gating mechanism can be functional, which also indicates that more than two parameters in each behavior are not supported in GLIF. Plus, these extra parameters function as bio-features that are verified workable in prior works. SRM offers more parameters. That is true, but the additional parameters are involved in some temporal convolution operations. For one thing, such an operation is very computationally costly. For another, this operation does not prove effective in improving SNNs, (it could be effective but not proven to be more effective than normal LIFs in deep SNNs). That is why we do not consider a unified gated SRM model. But, this could be a very interesting future research. Essentially, this work is based on LIF models as the title indicates.
> >
> > 4. We think the ultimate goal is to apply a higher biologically plausible model to SNN. To simulate such a spiking neuron, more parameters and more special operations are needed. Such as the Izhikevich neuron model and the H-H model. But, as we all know, the main obstacles to applying these models are the high computational cost and the low SNN performance. The proposed GLIF is able to formulate a spiking neuron with a higher representation space compared to vanilla LIF and its variants, and meanwhile improve the performance. This could be viewed as a trade-off between computation cost and performance. This is also a step to effectively fusing different bio-features
> > to shed light on the further study of complex neurons in SNNs.
> >
> > Above all, we admit your concerns about " why not we add more parameters? How many more is the best?" and the lack of justification are critical. But, in this work, our GLIF, as a plug-in method for SNNs unifying all these LIF variants and effectively improving SNNs, could be a positive answer to the issue "could the higher neuronal dynamic diversity of spiking neurons help SNNs do better?".
> >
> > Additionally, despite the concern of "weakness -2, -3, and -4 ", we think your mainly remaining concern is the "weakness -1", which is further explained above. We hope we could have further study to address your "weakness -1" in the future. And, hopefully, this could improve your rating a bit more.

---

### Official Review · Reviewer_ZaJt · 2022-07-26

**Rating:** 6
**Confidence:** 4
**Soundness:** 3 good
**Presentation:** 2 fair
**Contribution:** 4 excellent

**Summary:**

The authors propose Gated LIF, a heterogeneous spiking neuron, to fuse different bio-features in different neuronal behaviors, enlarging the representation space of spiking neurons. They use gating factors to determine the proportion of the fused bio-features, which are learnable during training. The authors show that the combination of different learnable membrane-related parameters and methods can make spiking neurons constantly changing, thus increasing the heterogeneity and adaptivity of spiking neurons.

**Questions:**

As GLIF neurons add more gated functionalities to the standard LIF neurons, it would be great to see the computational cost and latency change due to the added computational overload.

The results shown in Tables 4-6 are very confusing -
1. The results do not have any confidence interval - does that indicate that this is the best performance observed?
2. The results shown in Table 1 and Table 4 does not match - table 4 says the average accuracies are much higher than the accuracies on the same architecture and dataset shown in Table 4
3. Why did the authors choose T=4 for Table 4,5,6 when Table 1 clearly shows that T=6 consistently gives better results.

The authors in the ablation studies show the "Effectiveness of different bio-features." and "Importance of learnable gating factors" as separate sections. It would be great if the authors could discuss the difference between these two ablation studies and why they are essential.

In the ablation studies, the parameter sharing scheme is a bit hard to follow, and it would be great if the authors could add a bit more context and/or some references for the same.

**Limitations:**

The authors did not discuss the limitations of their work or the potential negative social impact of their work in the paper. The authors can discuss how the use of GLIF can make the SNN more biologically plausible, and by understanding the nature these SNN works, we can make more energy-efficient neural network models.


**Strengths And Weaknesses:**

Strengths:
 GLIF controls the fusion of different bio-features using different gating units for those three neuronal behaviors. GLIF can simultaneously contain different neuronal dynamic features, increasing the neuronal dynamic diversity of LIF models.

The GLIF model consistently improves the performance of all the datasets and the models. An interesting thing to note is that GLIF consistently achieves 100% accuracy with a 0% error margin on the DVS Gesture dataset, which is impressive.




Weakness:
I felt the paper is generally a bit hard to read and very dense. Also, the structure of the paper's organization makes it confusing for readers unfamiliar with spiking neural networks. As such, it would be greatly appreciated if the authors could elaborate on the idea and make the contributions of the work a bit more explicit.

The author uses some terminology without explaining what it means, as it might be new for people from different backgrounds. For example, the main premise of the paper is based on the hypothesis that GLIF fuses different "bio-features" in different neuronal behaviors, expanding the representation space of spiking neurons. However, it is difficult to understand what the authors mean by "bio-features" as it has not been defined anywhere in the manuscript.
The author discusses the variation of the neuronal dynamics using Fig. 2 and Section 3.3, which seemed a bit redundant and can be moved to the Appendix.

---

> ### Author Response · Authors · 2022-08-02
> **Response to Reviewer ZaJt:**
>
> Thank you for your thorough feedback. We first list your advice and questions, then give our detailed answers.
>
> > Adv. 1: It would be greatly appreciated if the authors could elaborate on the idea and make the contributions of the work a bit more explicit.
>
> A. 1:  We partially rewrite our introduction in our revised paper to elaborate on the idea of GLIF (**Line 38-56**) and summarize our contributions (**Line 57-65**). For one thing, GLIF is proposed to enlarge the representation space. For another, with our channel-wise parameteric method, SNNs with higher neuronal dynamic diversity can be built. Therefore, we can further figure out the issue: could higher neuronal dynamic diversity of spiking neurons help SNNs do better?.
> > Adv. 2: it is difficult to understand what the authors mean by "bio-features" as it has not been defined anywhere in the manuscript.  The author discusses the variation of the neuronal dynamics using Fig. 2 and Section 3.3, which seemed a bit redundant and can be moved to the Appendix.
>
> A. 2: The explanation of "bio-features" is described in the revised version **Line 123 to Line 125**.  Since the membrane potential traces over time can explicitly depict the neuronal dynamics, we mean to use a simple case to analyze the dynamic properties of GLIF, as illustrated in Fig. 2 and Sec. 3.3.
> > Adv.3: the parameter sharing scheme is a bit hard to follow, and it would be great if the authors could add a bit more context and/or some references for the same.
>
> A. 3: We add more detailed description of the parameter sharing scheme to the revised version of our paper,  "Channel-wise parametric method", Section 3.2.  Usually, spiking neurons in the same layer share the same values of the membrane-related parameters. Such parameter sharing scheme is called the layer-wise sharing scheme.
> > Adv.4: The authors did not discuss the limitations of their work or the potential negative social impact of their work in the paper. The authors can discuss how the use of GLIF can make the SNN more biologically plausible, and by understanding the nature these SNN works, we can make more energy-efficient neural network models.
>
> A. 4:  In this work, we investigate the hybrid bio-features in LIF models to figure out the issue: could higher neuronal dynamic diversity of spiking neurons help SNNs do better? Consequently, our GLIF offers a plug-in    method to increase the neuronal dynamic diversity of spiking neurons and improve the performance of SNNs. The limitation of our work mainly lies in the extra computation cost for GLIF. Our response to this common concern is provided in General Response.  Mining the biological plausibility and researching the nature of GLIF could be a valuable revelation for further study.
> > Q. 1: The results shown in Tables 4-6 are very confusing -The results do not have any confidence interval - does that indicate that this is the best performance observed?
>
> A. 5: Yes, we report the best performance observed as most of prior works do. Since SNNs training is very time consuming, we only run one time for each experiment on Ablation Study under the same random seed.
> > Q. 2: The results shown in Tables 4-6 are very confusing -The results shown in Table 1 and Table 4 does not match - table 4 says the average accuracies are much higher than the accuracies on the same architecture and dataset shown in Table 4.
>
> A. 5: Results in Tables 4-6 are one-time runs. Results in Table 1. are run three times under different random seeds following previous work, and we report "mean $\pm$ std. ". Furthermore, the supervised direct learning of SNNs has not been solid enough on this field by far. As listed in Table 1, both TET and DSpike are dedicated to improve the learing algorithm of SNNs and propose different training techniques, but our GLIF-based SNNs only adopt the bald learning algorithm without any sophisticated learning tricks and surpass their performance. The inadequate training of SNNs and the statistically weak confidence interval may account for the unmatched experimental results in Table 1 and Table 4. Nevertheless, results in Table 1 and Table 4 still demonstrate the effectiveness of our proposed method.
> >  Q. 3: The results shown in Tables 4-6 are very confusing - Why did the authors choose T=4 for Table 4,5,6 when Table 1 clearly shows that T=6 consistently gives better results.
>
> A. 6: On the one hand, as shown in Table 1 and Table 6, the performance of SNNs increases as the TimeStep increments under different experimental settings. On the other hand, although, SNNs with TimeStep 6 achieve better results. Their training process is quite time consuming since we adopt the bold learning algorithm. Training SNNs with TimeStep 6 on an A100 GPU for 200 epochs will cost around 1.5 days. Therefore, we choose SNNs with TimeStep 4 as a trade off between performance and time cost for our Ablation Study.

---

### Author Response · Authors · 2022-08-02
**General Response**

We very appreciate all reviewers for your meaningful feedback and reasonable concerns.  Here we would like to address common concerns and critical issue in General Response.

> Bug issue:  **On DVS128 Gesture** benchmark, we take train set as the test set and gets 100% accuracy. From reviewer jY7N.

In experiments of event-based dataset DVS128 Gesture, we made a very rookie but severe mistake that we take train set as the test set.  After the notice of such a mistake, we immediately fixed that bug and re-run the experiments on DVS128 Gesture.  However, we observe that the experimental results are quite unstable.  We run under different random seeds with GLIF-based 7B-Net [1]: 94.79, 93.40, 91.32, and 96.88.  And, we tried to reproduce the original 7B-Net [1] with their open source and instruments under different random seeds.  Worse still, we didn't achieve any results that over 95.00, which are much lower that the results reported in [1].  This is mainly because DVS128 Gesture is too small for it only contains 1176 training  samples and 288 test samples.

Therefore, we are convinced that DVS128 Gesture is **not** suitable for the testification of our methods.  As such, we validate our proposed GLIF on a much more bigger and high-confidence benchmark **CIFAR10-DVS**.  We still use the same architecture of [1], because this work is the most close to ours.  And, our GLIF-based SNN easily achieves a superior performance:

| Method    | Architecture | Timestep | Accuracy |
| --------- | ------------ | -------- | -------- |
| STBP-tdBN | ResNet-19    | 40       | 67.80    |
| LIAF      | LIAF-Net     | 10       | 70.40    |
| LIAF+TA   | TA-SNN-Net   | 10       | 72.00    |
| PLIF      | PLIF-Net     | 20       | 74.80    |
| SEW+PLIF  | 7B-wideNet   | 16       | 74.40    |
| GLIF      | 7B-wideNet    | 16       | 76.80    |


We have already revised our paper.  Because DVS dataset is not used in our ablation study or any other places.  After replacing DVS128 Gesture part with CIFAR10-DVS, our work remains complete and the validation becomes even more solid due to the high-confidence of CIFAR10-DVS. **The codes, the training logs and the trained model for CIFAR 10-DVS have already been uploaded to the Supplementary Material**.  The whole project will be open release.

> Computaional cost and latency. From reviewers ZaJt and i4xG.

The computational cost for single GLIF-based neuron is more than that of the vanilla LIF neuron.  However, in deep SNNs, the most compution cost is still on multiplication operations between weights and feature maps.  We did the following statistics on the compuational cost of CIFAR-10 task.

|Model  |Weight& FM (FLOPs/M) | LIF (FLOPs/M) | LIF/total% | GLIF(FLOPs/M) | GLIF/total% |
| -------- | ------------- | ---------- | ------------- | ----------- |---------- |
| ResNet-18            | 141.36         | 0.188      | 0.133         | 0.565       |0.398 |
| ResNet-19            | 2285.37       | 1.44       | 0.0630         | 4.33        |0.189|

Since GLIF offers a plug-in method to improve SNNs, such extra computational overhead, compared to the whole SNN computation, is negligible.

We also record the forward and backward latency of LIF-based and GLIF-based SNNs on single A100 GPU with a minibatch.

|          | LIF forward latency (s) | LIF backward latency (s) | LIF latency (s) | GLIF forward latency (s) | GLIF backward latency (s) | GLIF latency (s) |
| :------: | :---------------------: | :----------------------: | --------------- | ------------------------ | ------------------------- | ---------------- |
| Reset-18 |          0.25           |           0.08           | 0.33            | 0.58                     | 0.18                      | 0.77             |
| Reset-19 |          0.26           |           0.06           | 0.33            | 0.54                     | 0.20                      | 0.74             |

Such a  latency gap between LIF and GLIF SNNs is mainly caused by the poor support of PyTorch in handling channel-wise operation that GLIF requires.  Since the operation overhead incurred by GLIF is not that much, we think a GLIF tailored CUPY framework would greatly alleviate the latency.



[1] Fang W, Yu Z, Chen Y, et al. Deep residual learning in spiking neural networks[J]. Advances in Neural Information Processing Systems, 2021, 34: 21056-21069.

---

### Meta-Review · Area_Chair_VJvY · 2022-08-26

**Recommendation:** Accept
**Confidence:** Less certain

**Metareview:**

Although the scores are boarderline, the reviewers appreciate the proposed model as a novel as an interesting generalization of multiple existing models. The authors ironed out a bug in the evaluation, that was caught by a review, and report that results still hold (on a larger dataset).

**Award:**

No

---

### Decision · Program_Chairs · 2022-09-14

Accept